# A Study on the Synthesis, Curing Behavior and Flame Retardance of a Novel Flame Retardant Curing Agent for Epoxy Resin

**DOI:** 10.3390/polym14020245

**Published:** 2022-01-07

**Authors:** Yong Sun, Yongli Peng, Yajiao Zhang

**Affiliations:** College of Materials Science and Engineering, Wuhan Institute of Technology, Wuhan 430074, China; zayajaqa@163.com

**Keywords:** DOPO, flame retardant, curing agent, epoxy resin

## Abstract

In this work, a flame retardant curing agent (DOPO-MAC) composed of 9,10-dihydro-9-oxa-10-phosphaphenanthrene-10-oxide DOPO and methyl acrylamide (MAC) was synthesized successfully, and the structure of the compound was characterized by FT-IR and ^1^H-NMR. The non-isothermal kinetics of the epoxy resin/DOPO-MAC system with 1% phosphorus was studied by non-isothermal DSC method. The activation energy of the reaction (Ea), about 46 kJ/mol, was calculated by Kissinger and Ozawa method, indicating that the curing reaction was easy to carry out. The flame retardancy of the epoxy resin system was analyzed by vertical combustion test (UL94) and limiting oxygen index (LOI) test. The results showed that epoxy resin (EP) with 1% phosphorus successfully passed a UL-94 V-0 rating, and the LOI value increased along with the increasing of phosphorus content. It confirmed that DOPO-MAC possessed excellent flame retardance and higher curing reactivity. Moreover, the thermal stability of EP materials was also investigated by TGA. With the DOPO-MAC added, the residual mass of EP materials increased remarkably although the initial decomposition temperature decreased slightly.

## 1. Introduction

Epoxy resin (EP), phenolic resin and polyester resin are three thermosetting materials. Because of their excellent physical and mechanical properties, chemical resistance, electrical insulation, processing properties and good adhesion properties, they are widely used in coatings, adhesives, electronic and electrical components and other fields [1,2,3]. However, the flammability of EP limits its further application. Thus, it is necessary to impart EP with good flame retardance [4,5,6].

Flame retardance of EP can be improved by introducing flame retardant, including additive type and reactive type ones [7,8,9]. The additive flame retardant has the advantages of a simple adding method, convenient operation and easy industrialization [10,11,12]. However, due to the absence of interaction between the additive and the resin, the flame retardant is easy to permeate and migrate, which can deteriorate the mechanical properties and flame retardance of materials [13]. By contrast with additive type one, reactive type flame retardant can avoid damage to the mechanical property of EP and solve emigration of additive type ones from the matrix through chemical connection between flame retardants and epoxy resin [14,15,16]. The reactive flame-retardant method includes reactive flame-retardant EP monomer and reactive flame-retardant curing agent. Among halogen-free flame retardants, phosphorus-containing flame retardants are popular products thanks to high efficiency. The P-containing flame-retardant monomer contains cyclotriphosphazene-, phosphaphenanthrene-, phosphonate-and phosphate-based EP monomers. However, flame-retardant EP monomer is usually used with unmodified epoxy resin due to the lower epoxy value and the poor mechanical properties of cured epoxy resin.

Compared with flame-retardant EP monomer, flame retardant curing agents are easily prepared and flexibility, and these have become a hot research area [17,18]. Recently, some flame retardant curing agent, such as P-modified Schiff-base, anhydride, aliphatic amine and imidazole, have been prepared to flame retard EP resins. For instance, Huo et al. [19] synthesized a DOPO-modified Schiff base curing agent (PBI) from 9,10-dihydro-9-oxa-10-phosphaphenanthrene-10-oxide (DOPO), 2-aminobenzothiazole and 1H-Imidazole-4-carbaldehyde for EP. The EP modified with PBI showed good flame retardance (UL-94 V-0; LOI ≥ 36%) and heat resistance. Liang et al. [20] developed a P-containing anhydride curing agent (BPAODOPE) for EP, and the EP/BPAODOPE system passed a V-0 rating when the phosphorus content reached to 1.75 wt%. Zhao et al. [21] prepared a phosphaphenanthrene/triazine-containing anhydride curing agent (TDA) that showed higher flame retardance than BPAODOPE at the same phosphorus content, which was attributed to the synergism of triazine and phosphaphenanthrene.

In addition, aliphatic amine and imidazole-based flame retardant curing agents also have been widely studied. For example, Shao et al. [22] synthesized a functional aliphatic amine curing agent (named DPPEI) from diphenyl phosphorus oxide and polyethyleneimine for EP. When the content of DPPEI was 35%, the EP/DPPEI passed a UL-94 V-0 rating and had a LOI value of 29.8% as well as a significant reduction in total heat release and smoke release. Xu et al. [23] prepared an imidazole-based flame retardant curing agent (named IDOP) from imidazole that showed excellent flame retardance due to free radical trapping effect in gaseous phase. Xie et al. [24] synthesized a furfural-based flame retardant containing DOPO (MBF-DOPO). The LOI of the epoxy composite reached 32.9% (with the V-0 rating in UL-94 test). The results showed that MBF-DOPO promoted the carbonization of the epoxy matrix and effectively isolated the gas and heat transfer during the combustion process, thus improving the fire resistance of the epoxy thermosetting resin.

The purpose of this work is to design and prepare an efficient P-N co-effect flame retardant for EP, greatly reducing the shortcomings and expanding the application range of EP. Because DOPO is a very efficient flame retardant structure and aliphatic amine has high curing activity, the combination of the two units will prepare a flame retardant curing agent with high activity and flame retardant efficiency. In this paper, a DOPO-based flame retardant curing agent, named DOPO-MAC, was synthesized from DOPO and MAC by the addition reaction. Its structure was confirmed by FTIR and NMR. In addition, the curing behavior of DOPO-MAC was investigated by non-isothermal kinetics. Finally, the thermal stability and flame retardance of EP was analyzed.

## 2. Experimental Procedure

### 2.1. Materials

DOPO (purity > 98%) was purchased by Shandong Mingshan chemical company (Linyi, China); Methyl acrylamide (MAC) (98%) and N,N-Dimethylacetamide (DMF) (99.9%) were bought by Aladdin (Shanghai, China); Epoxy resin (E-51, the epoxy value of 0.51 mol/100 g) was provided by Wuxi resin factory (Wuxi, China).

### 2.2. Synthesis of DOPO-MAC

A total of 0.01 mol DOPO, 0.01 mol methyl acrylamide and 10 mL DMF (as the solvent) were firstly added into a three-necked flask with a thermometer, reflux device and magnetic stirring device, and kept stirring until completely dissolved. Afterwards, the mixtures continued to stir at 140 °C for 4–8 h under N_2_. At the end of the reaction, the final product was obtained through filtration, washed with water and dried at in a vacuum drying cabinet at 60 °C for 10 h. The synthetic route is shown in Figure 1.

### 2.3. Synthesis of Cured DOPO-MAC/EP Composites

Different proportions of epoxy resin and as-obtained DOPO-MAC were blended at 90 °C for 1 h. Accordingly, the mixtures were poured into the mold, and cured at 100 °C for 1.5 h, 115 °C for 5 h and 135 °C for 2 h. The pure EP materials were prepared at room temperature with MAC as the curing agent. The formula of EP materials with different contents of flame retardant was shown in Table 1.

### 2.4. Characterization

Fourier Transform Infrared Spectroscopy (FTIR): Fourier Transform Infrared spectrometer L1600301 manufactured by PerkinElmer was used in the test. The synthetic product was a powder solid, so it was appropriate to use the KBr pressure plate method. A small amount of sample was mixed with KBr to be lapping, repressed and tested. The operation parameters were set to 4 cm^−1^, each sample was scanned 4 times, and the scanning range was 4000–400 cm^−1^.

Nuclear Magnetic Resonance Spectroscopy (^1^H-NMR): An Avance 600 NMR spectrometer manufactured by Bruker (Switzerland) was used. Dimethyl sulfoxide (DMSO) was used as the solvent, while tetramethylsilane (TMS) was used as an internal standard.

DSC Test: A DSC200F3 differential scanning calorimeter manufactured by Netcom Scientific Instruments (Shanghai, China) Co., Ltd. was used in the non-isothermal DSC test. A few samples were put into the furnace in a high-purity N_2_ atmosphere, and the flow rate was kept at 20 mL/min. Al_2_O_3_ was the reference material, and the heating rates were set at 5 °C/min, 10 °C/min, 15 °C/min and 20 °C/min, respectively. The test temperature ranged from room temperature to 250 °C, and the sample dosage was about 8 mg.

TGA was recorded by Netzsch 2209F1 thermo-gravimetric analyzer (Germany) under nitrogen atmosphere from 30 °C to 700 °C. Samples of about 15 mg were used in each measurement and were placed in an open oven at a heating rate of 10 °C/min.

LOI test was conducted on a TTech-GBT2406-2 oxygen index meter in view of GB/T 2406 and UL 94 test was carried out by a TTech-GBT2408 vertical burning test on the basis of GB/T 2408. The relevant spline sizes were 130× 6.5 × 3.2 mm^3^ (LOI test) and 130 × 13 × 3.2 m^3^ (UL 94). The samples were held 10 cm over the burner and rapidly removed after exposure to outer flame for 10 s (V-1: The flame is extinguished within 30 s after two 10 s combustion tests. It cannot ignite the cotton under 30 cm. V-0: The flame is extinguished at 10 s after two 10 s combustion tests).

Tensile and flexural experiments were performed on a CMT4104 universal testing machine at a speed of 2 mm/min according to GB/T 1040.2-2006 and GB/T 9341-2008, respectively. ZBC1251 pendulum impact testing machine was adopted to study Charpy impact strength of materials according to GB/T 1843-2008. Additionally, the dumbbell-shaped specimens with thickness of 2 mm were applied in tensile tests, and the rectangular specimens with the size of 80 × 10 × 4 mm^3^ were adopted in three-point bending tests. The result was the average of five measurements.

## 3. Results and Discussion

### 3.1. FT-IR and ^1^H-NMR Characterization of DOPO-MAC

The infrared spectra of the vacuum-drying product were measured by FT-IR spectrometer. The infrared spectra were as showed in Figure 2. As seen in Figure 2, some typical characteristic peaks such as P-H (2438 and 2386 cm^−1^), P = O (1204 cm^−1^) and P-O (914 cm^−1^) appeared in DOPO spectra [25]. For DOPO-MAC, P = O and P-O peaks still existed, while the P-H stretching vibration peak disappeared, which indicated the reaction between DOPO and MAC [26]. The absorbance peaks at 3439 and 1664 cm^−1^ were N-H and C = O stretching vibration, respectively.

The structure of the resultant product was further investigated by ^1^H-NMR spectra, as shown in Figure 3. The peak at 7.21 ppm was attributed to N-H bond; the peaks at 1.2 and about 2.5–3 ppm belonged to hydrogen protons of alkanes; the peaks at 7.21–8 ppm were ascribed to the benzene ring structure [27]. The peak area was consistent with the number of corresponding hydrogen atoms.

### 3.2. Kinetic Study on DOPO-MAC/Epoxy Resin System

There are three hypothesis that should be followed when it comes to kinetic investigations [15]:

(1) The total area of the exothermic curve is directly proportional to the total heat released during the curing reaction.
(1)$$\frac{d\alpha }{dt}=\frac{dH}{dt}\times \frac{1}{\Delta H}$$
where Δ*H* represents the heat release of the whole curing reaction, *dH/dt* is the heat flow rate, d*α*/*dt* is the curing reaction rate.

(2) The reaction rate during curing is directly proportional to the heat flux at that time.
*dα/dt = k(T)f(α)*(2)
where *α* indicates the degree of curing reaction; *f*(*α*) is a function of *α*, and the specific form is determined by the curing mechanism; and *k(T)* is the reaction rate constant, which is determined by Arrhenius equation:(3)$$k(T)=A\times \exp(\frac{-Ea}{RT})$$
where *A* is the former factor, *E* the activation energy, *R* is gas constant and *T* is the temperature.

Based on the above assumptions, many model methods can be used to calculate the kinetic parameters of epoxy resin curing reaction, including the equations of Kissinger, Flynn-Wall-Ozawa and Crane. 

#### 3.2.1. Curing Reaction Heat

Epoxy resin system containing 1.0% of phosphorus was investigated by non-isothermal DSC tests with the heating rate of 5 °C/min, 10 °C/min, 15 °C/min and 20 °C/min. As shown in Figure 4, the initial curing temperature (T_i_), peak curing temperature (T_p_) and termination curing temperature (T_t_) of a non-isothermal DSC curve change due to changes in the heating rate. As shown in Figure 4, the curve of the curing reaction was smooth and there was only one significant exothermic peak. The graph showed the trend of the characteristic temperature during the curing process changing with the heating rate. It can be seen that the onset temperature, peak temperature and termination temperature of the curing reaction gradually increase as the heating rate (β) increases.

According to Table 2, a linear relationship between characteristic temperature and heating rate can be plotted, as shown in Figure 5. By plotting the T_i_, T_p_ and T_t_ with the heating rate β, and using an extrapolation method to make β = 0, the optimal curing temperature for static cure is obtained, which was T_i_ of 101 °C, T_p_ of 116 °C and T_t_ of 134 °C, respectively.

#### 3.2.2. Determination of Solidification Kinetic Parameters by Kissinger and Ozawa Method

The Kissinger method is one of the most commonly used curing kinetics analysis methods. By doing multiple scans on the cured sample at different heating rates and calculating the peak temperatures at different heating rates, the apparent activation energy of the curing reaction is obtained. The method assumes that the maximum reaction rate of the curing process occurs at the peak temperature, and differentiates and approximates the calculation equation of the curing reaction rate equation. The basic Equation is:(4)$$\ln\frac{\beta }{Tp\times Tp}=\ln\frac{A\times R}{Ea}-\frac{Ea}{R}\times \frac{1}{Tp}$$

As shown in Figure 6, the apparent activation energy of the solidification system was 45.11 kJ/mol, the A was 3.6 × 10^6^ /s, and the *n* was 0.93 from the slope and intercept of the straight line.

The Ozawa method is another method to obtain dynamic parameters whose advantage is that experimental errors caused by different assumptions about the reaction mechanism function can be avoided. It is usually used in checking the activation energy value obtained by other methods. The Ozawa method formula can be expressed as:(5)$$\ln\beta +1.0516\times \frac{Ea}{RT}=C$$

We obtained a straight line by plotting ln *β*—(1⁄Tp) (see Figure 7) and making linear regression. The *Ea* of 46.15 kJ/mol of the curing reaction can be obtained from the slope of the line.

The reaction activation energy Ea calculated according to the Kissinger method and the Ozawa method was approximately 46 kJ/mol, indicating that the reaction was easy to carry out.

### 3.3. Study on Flame Retardancy of Epoxy Resin System

LOI and UL-94 tests were carried out to investigate the flame retardancy, and the results are shown in Table 3. From Table 3, it can be seen that different phosphorus content of epoxy resin system could significantly increase the vertical combustion grade and limit oxygen index of the material as the content of phosphorus in the component increases, indicating that the introduction of curing agent DOPO-MAC greatly improved the flame-retardant effect of epoxy resin. When the phosphorus content was 1.0%, the P-1.0 system reached a UL94 V-0 with an LOI of 30.9%.

When the content of phosphorus was 1.5%, LOI could reach up to 31.8%, which indicated that with the increase of phosphorus content in the epoxy resin system, the flame retardant effect was obviously enhanced. The introduction of flame retardant curing agent named DOPO-MAC played a very important role in the epoxy resin system. The reason could be described as follows: When the resin system was heated and burning, the P-O-C bond of DOPO structure in the system would be broken and rearranged to form phosphoric anhydride or polyphosphoric acid products to dehydrate the polymer and reduce the ambient temperature. At the same time, the thermal decomposition of phosphoric acid could promote the carbonization of the polymer and form a compact carbon layer to isolate heat and oxygen, and prevent the release of combustible gases from the heat burning process of the polymer. The test results showed that the synthesized curing agent DOPO-MAC used in the curing of epoxy resin system could prepare high efficiency and halogen free flame retardant epoxy resin materials.

### 3.4. Thermal Properties

The thermal stability of the resin system was further studied by thermogravimetric analysis (TGA). As shown in Figure 8, the thermal weight loss curve of samples having phosphorus content from 0% to 1.5% for epoxy resin system was measured in the atmosphere of N_2_, the initial thermal decomposition temperature (T_5%_) of four different resin systems was recorded according to the thermal weight loss curve, and the maximum thermal decomposition temperature (T_max_) and the residual mass were recorded as shown in Table 4.

Where T_d, 5%_ was the temperature when mass loss weight was 5 wt%, T_d, max_ was the temperature when mass loss weight was 50%. Figure 8 and Table 4 show that the thermal decomposition of the resin system has only one weight loss interval in the atmosphere of N_2_, and with the increase of phosphorus content, the T_d, 5%_ and T_d, max_ of the resin system were reduced to a certain extent. The phosphorus containing groups easily decomposed at a relatively low temperature. Therefore, the introduction of phosphorus-containing structure had an adverse effect on the thermal stability of the solidified substance [28,29]. On the other hand, the P-O-C bond caused the initial decomposition of the polymer system due to thermal cracking in a lower temperature. With the increase of phosphorus, the residual mass increased, indicating that the introduction of DOPO-MAC increased charring ability [30,31].

Phosphorus-containing components decomposed and produced phosphoric acid, which acted as dehydrating agent to promote the dehydration of polymers into chars. The dense carbon layer inhibited the overflow of flammable gas, and isolated oxygen and heat source so as to prevent the polymer thermal oxidation, reduced the oxidation of heat and improved the flame retardant performance [32,33].

Figure 9 presents the tan δ and storage modulus curves, and the relevant data including glass transition temperature (T_g_) and crosslinking density (Ve) are collected in Table 4. From Table 4, the T_g_ of flame retardant epoxy resins showed a growing trend with the increasing of phosphorus content. The Ve of flame retardant epoxy resins declined compared with unmodified epoxy, which was attributed to large steric resistance of DOPO groups. In addition, the storage modulus of flame retardant epoxy resins at 50 °C were bigger than unmodified epoxy, suggested that rigidity of samples was improved. Thus, the improvement in T_g_ was mainly attributed to the greater rigidity and the limited movement of epoxy chain segments caused by DOPO groups.

### 3.5. Mechanical Properties

Figure 10 shows the tensile and flexural properties of all the cured resins. Both the tensile strength and modulus were increased with the addition of DOPO-MAC. P-0% system showed an average tensile strength of 74.3 MPa. For P-0.5%, P-1.0% and P-1.5%, their tensile strength was 75.4, 80.5 and 83.5 MPa, respectively. Their flexural performances presented Figure 10 also illustrated almost the same variation trend as the tensile properties. From P-0% to P-1.0%, a slightly increment in flexural strength was observed, which should be attributed to the compatibility and higher rigidity of systems containing DOPO-MAC.

## 4. Conclusions

In this paper, a novel flame retardant curing agent was synthesized, and used to cure and flame retard epoxy resins. The study on curing behavior showed that DOPO-MAC was an effective curing agent for epoxy resin. Additionally, the results of combustion test showed that both vertical combustion grade and the LOI value of the cured product increased with the increase of phosphorus content in the epoxy resin system. The LOI values of DOPO-MAC/epoxy resin were increased by 22.8% for P-0, 28.4% for P-0.5, 30.9% for P-1.0 and 31.8% for P-1.5, and the cured product reached UL 94 V-0 rating when the phosphorus content was 1.0%, which demonstrated the DOPO-MAC could effectively improve the flame retardant properties of epoxy resin materials. In addition, the composite materials had excellent mechanical properties due to their good compatibility and excellent structure.

## Figures and Tables

**Figure 1 polymers-14-00245-f001:**
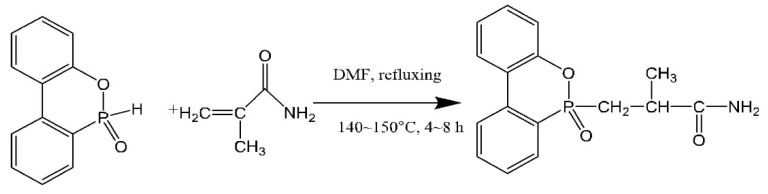
Synthetic route of DOPO-MAC.

**Figure 2 polymers-14-00245-f002:**
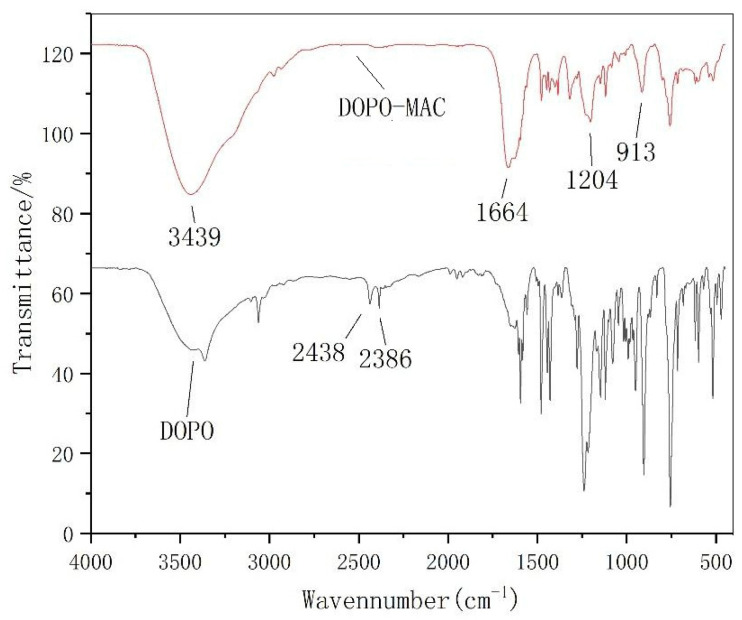
FT-IR of DOPO and DOPO-MAC.

**Figure 3 polymers-14-00245-f003:**
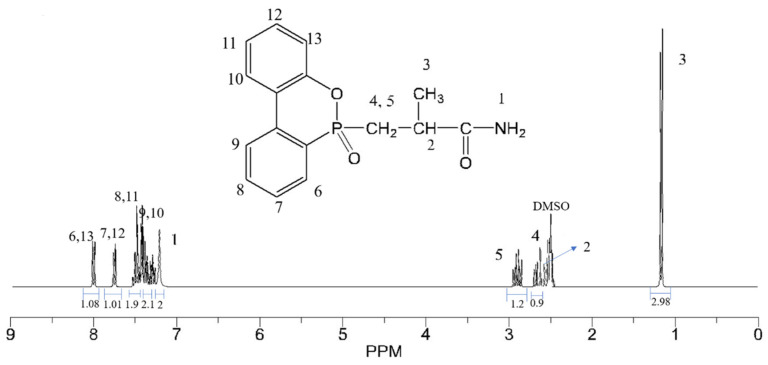
^1^H-NMR of DOPO-MAC.

**Figure 4 polymers-14-00245-f004:**
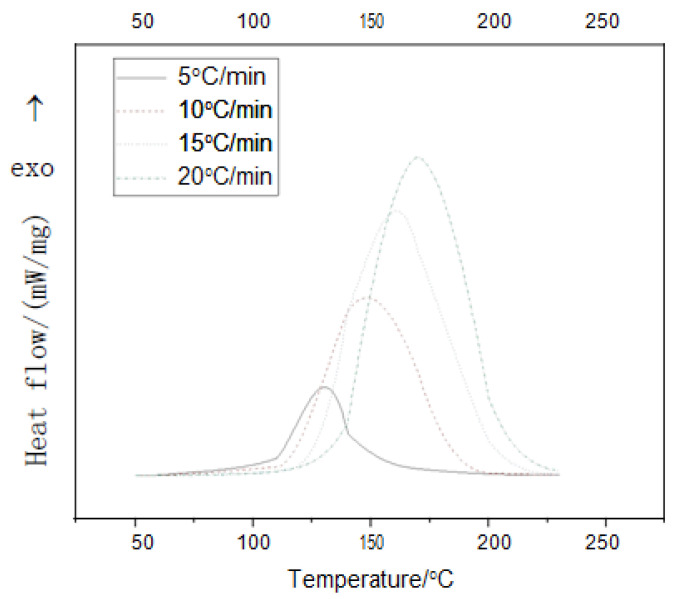
DSC curves of epoxy resin curing system at different heating rates.

**Figure 5 polymers-14-00245-f005:**
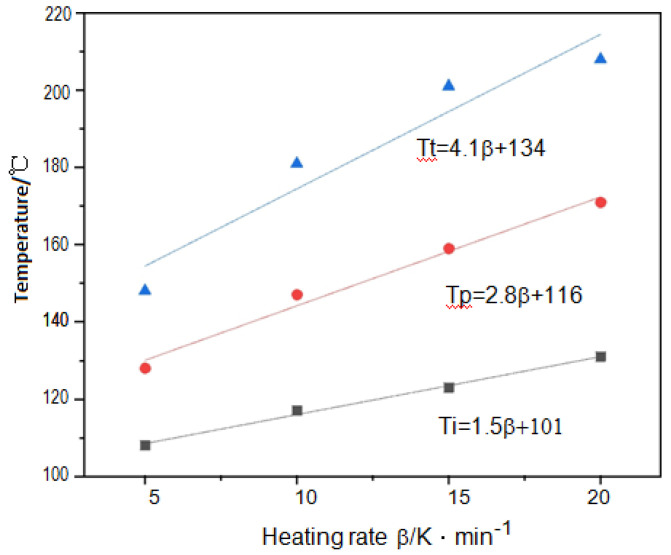
Relationship between β and T_i,_ T_p_ and T_t_ in curing system.

**Figure 6 polymers-14-00245-f006:**
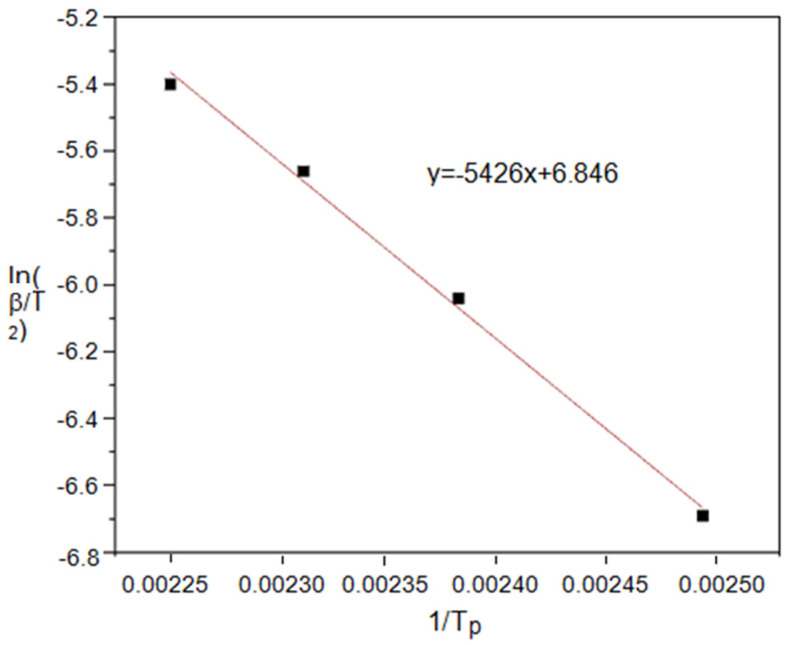
Linear regression equation between ln(β/T^2^) and 1/T_p_.

**Figure 7 polymers-14-00245-f007:**
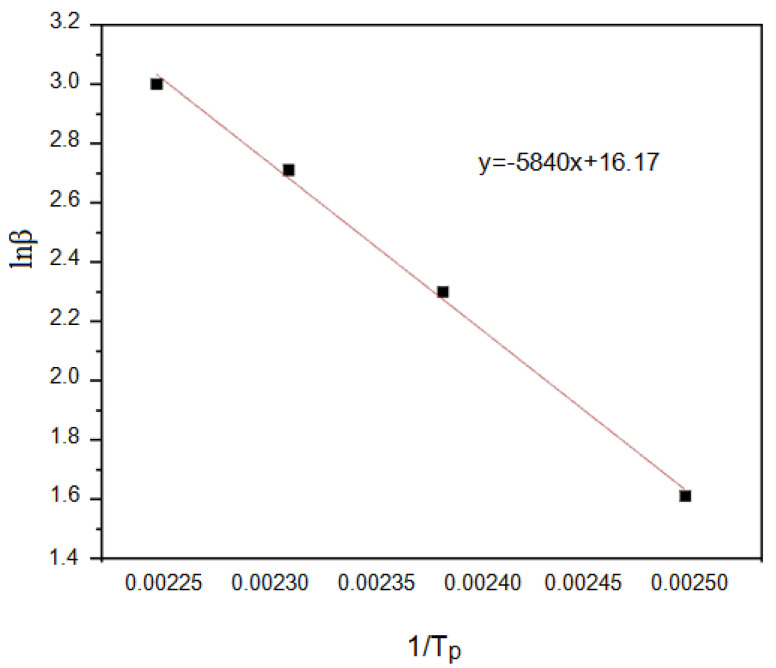
Linear regression equation between lnβ and 1/T_p_.

**Figure 8 polymers-14-00245-f008:**
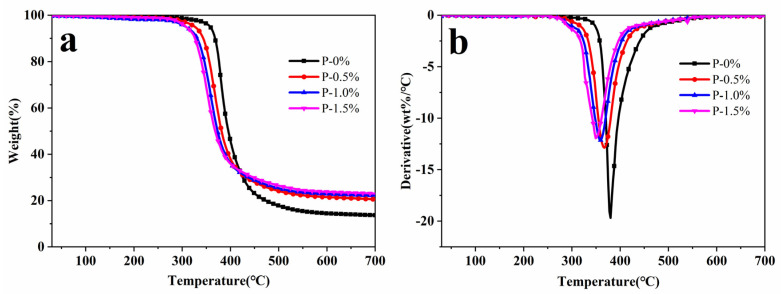
TGA (**a**) and DTG (**b**) curves of different resin systems in N_2_.

**Figure 9 polymers-14-00245-f009:**
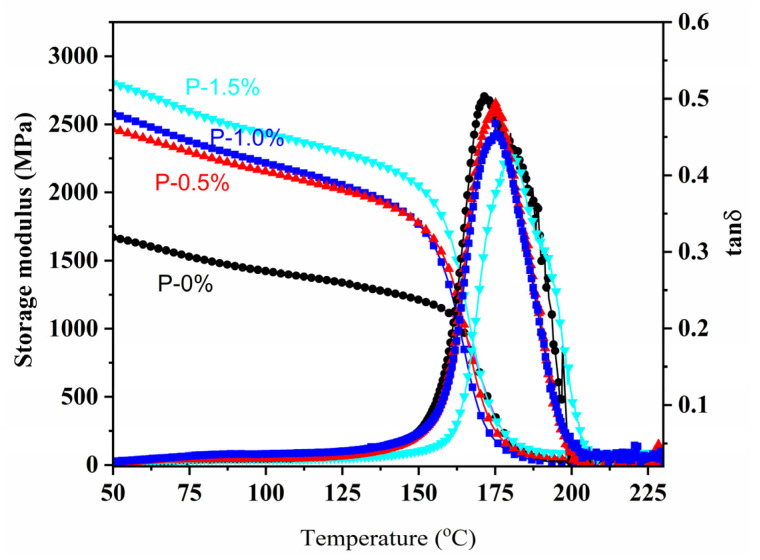
DMA curves of epoxy resins.

**Figure 10 polymers-14-00245-f010:**
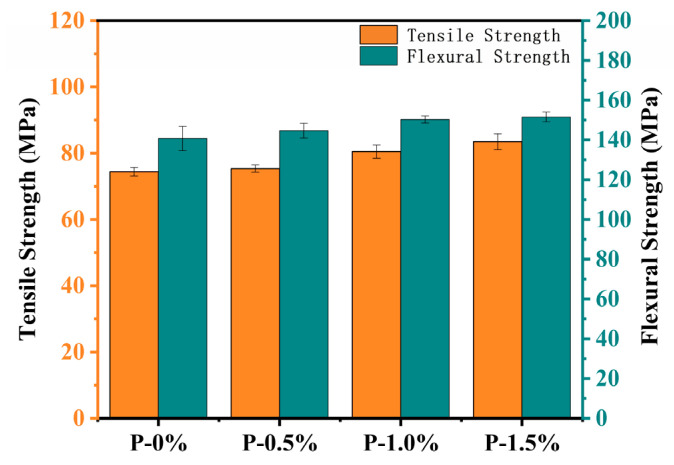
Tensile and flexural strengths of composites.

**Table 1 polymers-14-00245-t001:** The formula of EP materials.

Sample	EP (g)	MAC (g)	DOPO-MAC (g)	P Content (%) ^a^
P-0%	95	5	0	0
P-0.5%	95.2	0	4.8	0.5
P-1.0%	90.3	0	9.7	1
P-1.5%	85.5	0	14.5	1.5

^a^ P content = (m_1_ × 10.3%)/(m_1_ + m_2_); m_1_ is the mass of flame retardant; m_2_ is the mass of EP; 10.3% is the mass fraction of phosphorus in the flame retardant molecule.

**Table 2 polymers-14-00245-t002:** Corresponding data of DSC curves at different heating rates.

Heating Rate (K/min)	T_i_ (°C)	T_p_ (°C)	T_t_ (°C)
5	108	128	148
10	117	147	181
15	123	159	201
20	131	171	208

**Table 3 polymers-14-00245-t003:** Flame retardant properties of different resin systems.

Phosphorus (wt.%)	UL94	LOI (%)
0	No rate	22.8
0.5	V-1	25.4
1.0	V-0	28.9
1.5	V-0	31.8

**Table 4 polymers-14-00245-t004:** TGA and thermomechanical data.

Phosphorus (%)	T_d, 5%_ (°C)	T_d, max_ (°C)	Residue (wt%)	T_g_ ^a^ (°C)	Ve ^b^ (mol/m^3^)
0	350	380	11.2	175.0	2325
0.5	338	370	17.2	177.5	2200
1.0	326	360	20.7	178.0	2105
1.5	300	355	22.1	181.5	1984

^a^ Determined by DMA. **^b^** Crosslinking density, Ve= E’/3RT, E’ represents storage modulus at T (equals to T_g_ + 30 °C), R is the gas constant.

## Data Availability

The data presented in this study are available on request from the corresponding author.

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
