# Peer review of "A Study on the Synthesis, Curing Behavior and Flame Retardance of a Novel Flame Retardant Curing Agent for Epoxy Resin"

_polymers, 2022, doi:10.3390/polym14020245_

Round 1

Reviewer 1 Report

The article has been supplemented in accordance with the review presented before the rejection of the manuscript in the previous iteration (polymers-1278545). Still, the introduction is short, and, in my opinion, it does not entirely refer to the research that has been conducted so far. However, manuscript can be recommended for the editorial procedure. However, it requires significant editorial changes and corrections. The authors should carefully check the text before its final submission.

Author Response

Comment 1. The article has been supplemented in accordance with the review presented before the rejection of the manuscript in the previous iteration (polymers-1278545). Still, the introduction is short, and, in my opinion, it does not entirely refer to the research that has been conducted so far. However, manuscript can be recommended for the editorial procedure. However, it requires significant editorial changes and corrections. The authors should carefully check the text before its final submission.

Reply. Thank you for your good comments. We have revised the introduction according to your comments.

Reviewer 2 Report

You can find my observations written in blue in the attached document.

Author Response

Comment 2. You can find my observations written in blue in the attached document.

Reply. Thank you for your comments. We have revised the manuscript according to attached document.

This manuscript is a resubmission of an earlier submission. The following is a list of the peer review reports and author responses from that submission.

Round 1

Reviewer 1 Report

-The text does not present the development and chemical structure of DOPO

- Introduction should be significantly expanded. Considering that one of the main objectives of the work is to study the changes in the reactivity of the flame retardant modified epoxy resin, it is necessary to refer to the potential mechanisms and limitations that have been encountered and described in similar systems in the literature. Moreover, bearing in mind the extensive scope of work currently being carried out on novel flame retardants, the literature part emphasizing the innovation of the developed solution with others presented so far must be very strongly emphasized.

From the point of view of studying the kinetics of the thermosetting resins, a cross-linking process is essential in the course of the reaction and in the final effect in the form of cross-linking density. It must be determined by the Flory Rehner method or indirectly using thermomechanical analysis.

- Unfortunately, the presentation of new flame retardants without carrying out the most crucial test, which is measurements using a cone calorimeter or at least a microcalorimeter, does not allow for a complete description of the detachment efficiency compounds produced. In my opinion, without presenting a complete flammability analysis, the submitted test results constitute only a short communication, which, without supplementation, cannot be recommended as a complete manuscript that meets the publication criteria in the high impact journal, which is Polymers.

- I miss a mechanical characterization, how is the tensile strength of the material, how is the flame retardant polymer interaction

- From an editorial point of view, the manuscript is very sloppy and requires significant improvement. This affects not only the formatting of the text but also its quality.

- The characteristics of the materials, and in particular of the epoxy resin used, are insufficient.

- The characterization of the synthesized materials is not convincing. Moreover, the TG curve should be explained with the additional presentation of DTG, including all used materials, not only final samples.

  • An increase of LOI from 22.8 to 31.8% is not a story of success. Materials of 21 <LOI <24 must not be called “self-extinguishing”. Many materials show V-1 or V-0 self-extinguishing classifications when LOI> 27-29 vol .-% are reached. Some composites need LOI> 40 vol .-% before self-extinguishing in UL 94. The photography of samples during and after UL94 tests should be provided in the supplementary information.

Author Response

Thank you for your constructive suggestions. We have modified the introduction and introduced the corresponding reference reactive additives. Due to the lack of testing technology, we did not carry out the combustion test with cone calorimeter. The test cycle of the third-party organization is longer, generally more than three months. However, we only have a 10 day modification period, which can not support us to further supplement the data. We supplement the mechanical test and describe it in the paper. The composite materials had excellent mechanical properties. The characteristics of epoxy resin are supplemented in the material part. Finally, we invite professionals to modify the language

Reviewer 2 Report

My recommendation is that the manuscript can be published after minor revision, as resulted from the list below.

  1. The text contains many abbreviations that weren’t explained before they appear for the first time (including in Abstract section). For example: DOPO (r.6); EP (r.13); DPPEI (r.39); DMF (r.58)
  2. r. 14: “…with the increasing of phosphorus content.” There is no use to add in Abstract data about P-0, P-1.0…etc, because at this point the composition for these samples is unknown.
  3. r. 17: “…the residual mass of EP materials increased remarkably although…”
  4. r. 22: “Epoxy (EP), phenolic and polyester resins are called thermosetting materials. Because of their excellent…”
  5. r. 28: “…flame retardants, including…”
  6. r. 29: “By contrast…” this phrase is not very clear written. Please rephrase “avoid damage the mechanical properties”; “solve emigration of additive”
  7. r. 39: “…for researchers…”
  8. r. 44: “…(DOPO) that showed an excellent…”
  9. r. 48: “flame retardant are applied”?? or “high efficient”??? Please correct the English accordingly!
  10. r. 84: “…Switzerland was used.”
  11. r. 105: “…the reaction between DOPO and…”
  12. r. 109: “ FTIR spectra of…”
  13. r. 110: I recommend investigated instead of tested by NMR
  14. r. 118: “…three hypothesis that should be followed:”
  15. please supply the correlation parameter for the linear dependences of Tt, Tp and Ti.
  16. r. 167: please verify the value for parameter A!!!
  17. For Figures 6 and 7, there is no measure unit for 1/T, and also a correlation parameter should be provided for the linear dependences.
  18. r. 176: This is relation (5) not 7.
  19. Maybe the authors should provide a brief description of UL94 test results. What is V-1, V-0?

Author Response

Thank you for your scientific comments. We have corrected each of the questions you raised and marked the corrections in red. For the first abbreviation, we have added its full name. We have also corrected spelling and grammar errors in the article. The correlation coefficient of linear fitting is added to the paper. The parameter A value is corrected

Reviewer 3 Report

The manuscript titled, “A study on the synthesis, curing behavior and flame retardance of a novel flame retardant curing agent for epoxy resin” maybe an interesting work, however requires exhaustive revisions. My comments are below;

  1. The writing is very poor, very difficult to follow. Too many auxiliary words, such as besides, such as, so on… Too many self-claimed sentences, novel synthesis, it’s just modification. The authors should revise the article from a native English speaker or use professional editing service.
  2. The morphology must be discussed; SEM, and real photos of specimen are demanded.
  3. How FTIR could be verified for the occurrence of claimed novel flame retardant.
  4. The synergetic effect of phosphorus-nitrogen flame retardant should be discussed in introduction part. Use these two articles for references.
    1. https://doi.org/10.1142/S0218625X17501141
    2. https://dx.doi.org/10.22059/ijer.2016.57726
  5. The quality of the figures is too low, it needs to be improved.

Author Response

Thank you for your good suggestions. The papers are very excellent and helpful to our work, and have been cited in our revision manuscript. Due to the lack of testing technology, we did not study the morphology. The testing cycle of the third-party organization is longer, generally more than three months. However, we only have a 10-day revision period, which cannot support us to further supplement the data. Infrared spectrum can show specific groups. In the spectrum of the new flame retardant, there are characteristic peaks of benzene ring and P-O-P bond on DOPO, but no characteristic peaks of P-H, indicating that the P-H bond reacts to form new substances.

Reviewer 4 Report

Dear Authors,

Overall, this manuscript is interesting, but nevertheless has a few shortcomings that need to be corrected.
Detailed comments below:

Line 6: Each acronym should be properly described before first use.

Line 21: This introduction is rather sketchy. You have to refer to more literature.

Line 53: I think you need to better emphasize the purpose of the research.

Line 55: I admit I prefer: Materials and methods.

Line 57: Name (manufacturer, country, city). In general, all purchased materials should be described as such. This also applies to scientific equipment. Review the entire methodology in this respect.

Line 66: The dryer model needs to be added.

Line 98: There is a poorly guided discussion of the test results in the description of the test results. You must refer to a larger number of literature items.

Line 109: The description below the graph in Fig. 2 is very brief. It needs to be better developed. This also applies to Fig. 3.

Line 210: When describing the TGA results, there are no references to studies by other authors. Moreover, when describing the results, I think that it should be indicated how far the obtained results are from the desired material properties. Review the other descriptions of your research results for this as well.

Line 234. I propose to put the units on the x and y axes in the brackets. Adjust it on all charts.

Author Response

Thank you for your good suggestions. We have corrected each of the questions you raised and marked the corrections in red. For the first abbreviation, we have added its full name. In the introduction we emphasize the purpose of our research more. In response to your request, we have revised the method section. More literature is referred to in the introduction and results discussion section. We describe Fig. 2 and 3 more. We have also revised the table.

Round 2

Reviewer 1 Report

The authors responded laconically to selected comments and did not take into account the changes that were highlighted. 

The feedback on the 10-day review deadline is insufficient, because the Editor will certainly allow for the resubmission of the properly completed work. If the authors cannot supplement the data on the cross-link density by the solvent method, it is also possible to evaluate it indirectly by using the DMA analysis.

The authors did not refer to and supplemented, as recommended, the presented measurement data with the TGA analysis of modifiers, and did not compile the dTG curves. Photos of the samples were also not attached.

In my opinion, the comments made in the first review were downplayed and deliberately omitted by the authors. I leave further decisions for the Editor to evaluate, but in my opinion, the article should be significantly improved.

Author Response

I am sorry that the reviewer was not satisfied with the last reply. Indeed, due to the lack of experimental instruments, we have not been able to use more representational methods to explain more problems (such as cross-linking density testing by using the DMA analysis). Therefore, we have carried out research within our capacity on the basis of existing research. In this paper, we focus on the study of flame retardancy and curing kinetics of flame retardant curing agents by UL-94, limiting oxygen index and non-isothermal methods. The existing characterization in this paper can effectively illustrate the effectiveness of the flame retardant curing agent. In the future work, we are going to explore other properties of polymers with sufficient funds and instruments. About TGA test, some DTG data were also showed in Table 4, such as Tmax.

Reviewer 3 Report

No further comments to add.

Author Response

Thanks for the reviewer's constructive comments. We have done our best to polish the article and checked the grammar and spelling

Reviewer 4 Report

Dear Authors,

I accept the corrections made.

Author Response

We have made structural adjustments and supplements to the introduction part to better express the purpose of our research. We consider our research worthwhile.